# Fairness without Demographics through Knowledge Distillation

**Junyi Chai, Taeuk Jang, Xiaoqian Wang***
Elmore Family School of Electrical and Computer Engineering
Purdue University
West Lafayette, IN 47906
{chai28,jang141,joywang}@purdue.edu

## Abstract

Most of existing work on fairness assumes available demographic information in the training set. In practice, due to legal or privacy concerns, when demographic information is not available in the training set, it is crucial to find alternative objectives to ensure fairness. Existing work on fairness without demographics follows Rawlsian Max-Min fairness objectives. However, such constraints could be too strict to improve group fairness, and could lead to a great decrease in accuracy. In light of these limitations, in this paper, we propose to solve the problem from a new perspective, i.e., through knowledge distillation. Our method uses soft label from an overfitted teacher model as an alternative, and we show from preliminary experiments that soft labelling is beneficial for improving fairness. We analyze theoretically the fairness of our method, and we show that our method can be treated as an error-based reweighing. Experimental results on three datasets show that our method outperforms state-of-the-art alternatives, with notable improvements in group fairness and with relatively small decrease in accuracy.

## 1 Introduction

Machine learning has been widely adopted in real-world scenarios that has great social influence, and fairness in automatic decision-making systems has become an arising concern. It has been shown that without proper intervention, models can be biased against certain demographic groups (Hardt et al., 2016; Chouldechova, 2017; Ntoutsi et al., 2020). Due to distributional disparities across different demographic groups, simply removing sensitive attributes during training does not improve fairness. Therefore, research on fair classification has received extensive attention, and different methods have been proposed to improve fairness (Roh et al., 2020a; Choi et al., 2020). These methods usually focus on specified fairness notions, and most of these works assume access to predefined sensitive information during training.

In practice, however, due to legal or regulatory concerns, it is often infeasible to collect sensitive information, which greatly limits the usage of conventional methods on fairness. Besides, in many real-world applications, we expect the decision-making systems to be fair w.r.t. multiple sensitive attributes. For example, Consumer Financial Protection Bureau (CFPB) requests creditors to achieve fairness, without requesting or collecting information about an applicant's race, color, religion, national origin, or sex. As most works on fairness assumes only one predefined sensitive attribute, due to difference in distributional disparities across different sensitive attributes, improving fairness under one sensitive attribute does not always imply fairness under another sensitive attribute. Therefore, it is crucial to study the problem of fairness without demographics.

---

*Corresponding author.

36th Conference on Neural Information Processing Systems (NeurIPS 2022).

Current methods on fairness without demographics can be divided into two categories: fairness with proxy sensitive attribute and Max-Min fairness. Methods on fairness with proxy sensitive attribute (Yan et al., 2020; Grari et al., 2021) utilizes surrogate group information, including clustering information and group information through estimation as alternative, and enforces fairness under surrogate group partition. The problem with proxy attributes is that it requires a strong assumption that the clustering or surrogate group information is correlated with the sensitive attribute(s) of concern, and improving group fairness w.r.t. several sensitive arrtibutes could be infeasible under the condition of large distributional discrepancy over different sensitive attributes of interest. Methods on Max-Min fairness (Hashimoto et al., 2018; Lahoti et al., 2020) focuses on improving the worst utility amongst all sensitive groups. Ideally, this formulation is in accord with group fairness; however, in practice, such formulation could be too strict to obtain improvement in group fairness, as improving worst-group utility does not necessarily imply decrease in inter-group disparities, and since Max-Min fairness tends to require similar utility over all groups, this could lead to a significant decrease in accuracy. Instead, we seek to find alternative ways to quantify disparities without observing sensitive information during training.

We draw inspiration from debiased learning (Choi et al., 2019; Nam et al., 2020; Liu et al., 2021), where the goal is to unlearn spurious correlations present in training set. One common way of debiased learning is through reweighing, where samples of worst groups specified by spurious features are up-weighed so as to mitigate biased prediction. For classification tasks, one simple way to assign weight is to replace the discrete one-hot encoding with continuous labelling that are obtained based on expected properties. However, for most classification tasks, continuous labelling is unrealistic in real-world scenarios. In light of this, we consider obtaining the soft labels through knowledge distillation, where training labels for student model are replaced with normalized logits of teacher model, and we expect knowledge from the overfitted teacher model to help fit hard samples in training set.

In this paper, we propose a knowledge distillation framework for improving fairness without accessing sensitive information. Compared with previous works on fairness without demographics, our method does not impose surrogate constraints on fairness. Instead, we try to find a soft labelling for samples through knowledge distillation to help the student model better fit. During training, we first train a teacher network to be overfitted to training data, and we use the normalized logits of teacher model as soft label to train a student network. In this way, we seek to correct bias in training set via reweighing training samples.

We summarize our contribution as follows:

1. **Fairness without Demographics**: We study the effectiveness of knowledge distillation in improving fairness without observing sensitive information. We discuss two different mapping functions of logits by teacher model, and we discuss theoretically the impact of label smoothing and knowledge distillation on fairness.

2. Our method effectively utilizes knowledge from complex teacher model and achieves comparable classification results than unconstrained baseline student model.

3. **Empirical Observation**: We validate the effectiveness of label smoothing on new Adult dataset, and we show from experiments that our method achieves remarkable improvement in fairness on three benchmark datasets under different sensitive attributes.

## 2 Related Work

### 2.1 Group Fairness in Classification

Group fairness is a class of definitions to quantify fairness, which measures the disparity of predicted outcome among the subgroups with certain sensitive attributes. Several works were proposed to mitigate the disparity and ensure the independence of the performance measures between the subgroups to achieve group fairness. Equal opportunity (Hardt et al., 2016) states that true positive rates are the same for the subgroups. Similarly, predictive equality (Chouldechova, 2017) requires the equality of false positive rates. Applying calibration conditions (Kleinberg et al., 2016; Pleiss et al., 2017) were proposed so that the probability estimate matches the actual ratio of positive distribution within the group. Diverse approaches are presented to meet particular group fairness notions. Pre-processing method (Chen et al., 2018; Jang et al., 2021) suggests improving the skewed sample size

problem. Adversarial learning (Madras et al., 2018) is proposed to learn a fair representation that is independent of sensitive attributes. In-processing methods (Iosifidis and Ntoutsi, 2019; Roh et al., 2020b; Chai and Wang, 2022) aim at regulating the training process with fairness constraints by reweighing training samples or enforcing fairness regularization. Post-processing the output of the biased model (Kim et al., 2020; Jang et al., 2022) with multiple fairness objectives are introduced as they are more efficient than training a model from scratch. However, to ensure fairness, the aforementioned approaches require sensitive information such as race, gender, etc. This restricts their versatility because the sensitive information is not always available owing to regulations and privacy concerns. Work including (Celis et al., 2021a,b; Mehrotra and Celis, 2021) studies the problem of fair classification under noisy or adversarially perturbed sensitive attributes, which can be considered as a relaxed version of fairness without demographics.

## 2.2   Fairness without Demographics

Due to the practicality and regulatory limitations, methods to achieve fairness without demographics have emerged. The methods in this category adhere to Rawlsian Max-Min fairness (Rawls, 2009): minimize the risk of the group with the highest risk. Distributionally Robust Optimization (DRO) (Hashimoto et al., 2018) utilizes $\chi^2$-divergence to find the worst-case distribution that is close to the empirical distribution and minimizes the risk of such distribution. ARL (Lahoti et al., 2020) conduct weighted empirical risk minimization with adversary network to maximize the loss with the perspective of computationally-identifiability. Other methods (Zhao et al., 2022) on fairness without demographics seek to find surrogate group information based on input features. Fair class balancing (Yan et al., 2020) proposes to quantify disparities w.r.t. sensitive attribute using clustering information and upsample minor class in each cluster to achieve balance.

## 2.3   Debiased learning

Debiased learning focuses on learning representations that are unbiased w.r.t. spurious features. Some works on debiased learning focus on learning unbiased representation through adversarial training. Choi et al. (2019) propose to augment the standard cross-entropy loss for action classification with an adversarial loss for scene types and a human mask confusion loss for background. Kim et al. (2019) proposes to unlearn the bias information by minimizing mutual information between latent representation and bias prediction with one adversarial network. Another approach employs a biased model to explore the samples with high loss. Nam et al. (2020) proposed to apply GCE loss (Zhang and Sabuncu, 2018) to train a bias amplifying classifier and parallelly train a fair classifier by referencing the biased classifier. Liu et al. (2021) suggest a simple two-stage training process. After training a classifier with ERM in the first stage, during the second stage, the model emphasizes the misclassified samples in the first stage. However, the objective to minimize the maximum loss could make the training unstable and deteriorate the overall performance.

## 2.4   Label Augmentation for Classification

Unlike the previous works, we explore a new approach to ensure fairness without demographics, which is to utilize knowledge distillation (Hinton et al., 2015). While cross entropy (Rumelhart et al., 1986) has been a dominant loss function for classification tasks, some works question its shortcomings, such as being sensitive to noisy label (Zhang and Sabuncu, 2018), poor margin (Nar et al., 2018), and vulnerable to adversarial samples (Nar et al., 2019). To overcome this problem, perturbing the label distribution empirically verified the improvement. Label smoothing (Müller et al., 2019) and label augmentation (Zhang et al., 2017) effectively deal with noisy label and adversarial samples. Knowledge distillation (Hinton et al., 2015) was proposed for model compression to effectively transfer knowledge from a deep teacher model to a smaller student model. Surprisingly, by convex combination of classical cross entropy loss and supervision from the teacher model, the student model could obtain a competitive or even a superior performance at inference. Even though a family of knowledge distillation methods is proposed, its impact on fairness is not actively studied yet. In this work, we study the effectiveness of knowledge distillation on fairness in the absence of sensitive information, and we claim that it noticeably improves fairness by *distilling* intrinsic information while *volatilizing* data bias.

## 3 Method

### 3.1 Background

Consider labelled samples $(x, y, a) \sim \mathcal{X} \times \mathcal{Y} \times \mathcal{A}$, with $y$ being the ground truth labels and $a$ being the sensitive attributes, the goal of fair classification is to learn a classifier that satisfies certain fairness criteria w.r.t. $a$. These criteria can generally be formulated as independence between sensitive information and expected statistical features. For example, disparate impact (Feldman et al., 2015) requires the predicted label $\hat{y}$ to be independent of sensitive attributes: $\hat{y} \perp\!\!\!\perp a$, and equalized odds is satisfied when the predictions and sensitive attributes are independent conditional on ground truth label: $\hat{y} \perp\!\!\!\perp a | y$.

### 3.2 Preliminary Results

Conventional methods on fairness fails when no sensitive information is available in the training set. Instead, we seek to find surrogate training objective that is beneficial for improving fairness. We start with preliminary results on new Adult dataset.

The new Adult dataset (Ding et al., 2021) is a reconstructed version of Adult dataset (Dua and Graff, 2017). Instead of containing binary labels indicating whether or not an tindividual's income exceeds $50K$, the new Adult dataset provides specific incomes for each individual, and the goal is to predict whether an individual's income exceeds certain predefined threshold. Research on new Adult dataset shows that by adjusting the income threshold for binary encoding we are able to improve fairness, without introducing extra regularization terms during training (Ding et al., 2021). Intuitively, a properly adjusted threshold for binary labelling helps reduce hard samples in dataset, which could be beneficial for the unprivileged subgroup. Inspired by this, we consider reweighing samples through label smoothing. Although labelling bias have been widely studied in fairness literature (Jiang and Nachum, 2020; Krasanakis et al., 2018), the effect of label smoothing without accessing sensitive information remains unclear. *Does label smoothing also help improve fairness during training?*

We validate this hypothesis on new Adult dataset. Specifically, we repeat experiments three times and report average results, and in each repetition, we randomly split data into $50\%$ training data and $50\%$ testing data. Following the traditional setting on the Adult data, we set the binary cutoff threshold of income as $\xi = 50K$. Let $s_i$ be the income of $i$-th individual, on testing data, we have $y_i = \mathbb{1}(s_i > \xi)$, and we consider four different types of labelling during training:

- **Binary**: The training labels are determined by $y_i = \mathbb{1}(s_i > \xi)$.

- **Random**: The training labels are determined as follows: for $s_i > \xi$, we sample $y_i$ from a uniform distribution on $(0.5, 1]$; for $s_i \leq \xi$, we sample $y_i$ from a uniform distribution on $[0, 0.5]$.

- **Softmax**: The training labels are determined by $y_i = (1 + e^{\frac{s_i - \xi}{\xi}})^{-1}$.

- **Linear**: The training labels are determined by a piecewise linear function: $y_i = \frac{(s_i - \min_i s_i)}{2(\xi - \min_i s_i)} \mathbb{1}(s_i \leq \xi) + \left( \frac{(s_i - \xi)}{2(\max_i s_i - \xi)} + 0.5 \right) \mathbb{1}(s_i > \xi)$.

| Label Assignment | Accuracy | Sensitive Attribute: Race | | Sensitive Attribute: Gender | |
| --- | --- | --- | --- | --- | --- |
| | | Dis. Impact | Eq. Odds | Dis. Impact | Eq. Odds |
| Binary | **85.19±0.17%** | 11.80±0.48% | 13.22±1.70% | 17.74±0.56% | 16.65±1.80% |
| Random | 85.13±0.17% | 12.21±1.37% | 13.15±1.12% | 17.64±0.95% | 15.57±1.49% |
| Linear | 85.14±0.25% | **10.75±1.15%** | **10.13±1.06%** | 16.62±0.83% | **12.37±1.64%** |
| Softmax | **85.19±0.17%** | 11.21±0.74% | 10.67±1.24% | **16.37±0.58%** | 13.34±1.89% |

Table 1: Experimental results on new Adult dataset with race and gender as sensitive attribute, respectively. Fairness is evaluated using two metrics: Disparate Impact and Equalized Odds.

Results are shown in Tab. 1. Compared with binary labelling, both linear and softmax labelling improves fairness under different sensitive attributes, while random labelling achieves little improvement in fairness. This shows the effectiveness of proper label smoothing.

## 3.3 Knowledge Distillation

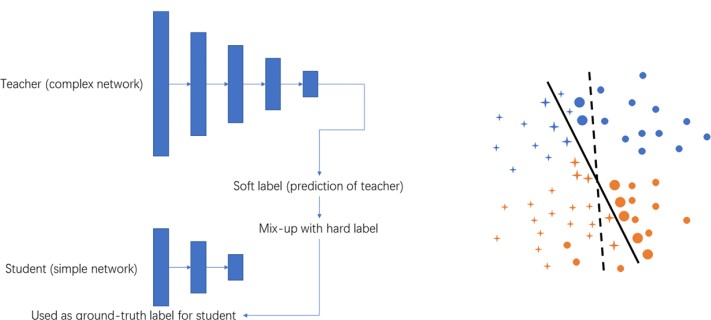

Figure 1: Demonstration of our knowledge distillation method. The dashed line refers to the decision boundary under soft labelling, and the solid line refers to the decision boundary under hard labelling. Points of different shapes refer to samples of different labels, different colors refer to different sensitive groups and larger pattern indicates higher weight. By focusing on hard samples that are correctly classified (marginal samples that are correctly classified), our method learns a better decision boundary which improves the performance on hard samples, and disadvantaged group(s) are more likely to benefit from this.

For most classification tasks with categorical encoding, continuous labels are not available. In light of this, we instead consider knowledge distillation for obtaining soft labels. We considering the binary classification setting, although our method may be readily generalized to multi-class classification.

Knowledge distillation is a form of model compression through the process of transferring the knowledge from a large pre-trained teacher model or set of models to a single smaller student model that can be practically deployed under real-world scenarios, and is performed commonly on neural networks associated with complex architectures. We consider response-based knowledge, which focuses on the final output layer of the teacher model, and the student model learns to mimic the predictions of the teacher model.

Generally, given a training set $\{(x_i, y_i), 1 \leq i \leq N\} \sim \mathcal{X} \times \mathcal{Y}$, let $g : \mathcal{X} \to \mathcal{Y}, x \mapsto \hat{y}^{\text{t}}$ be the teacher model, and $f : \mathcal{X} \to \mathcal{Y}, x \mapsto \hat{y}^{\text{s}}$ be the student model, the training objective for student model can be formulated as

$$L = \frac{1}{N} \sum_{i=1}^{N} \left[ \alpha L_{cls}(f(x_i), \hat{y}_i^{\text{t}}) + (1 - \alpha) L_{cls}(f(x_i), y_i) \right], \tag{1}$$

where $\hat{y}_i^{\text{t}} = g(x_i)$ is the soft label from teacher model, $L_{cls}$ is the classification loss, and $\alpha \in (0, 1)$ is the trade-off hyper-parameter. For a classification task with $c$ classes, the teacher model can be decomposed into two functions, $g = \phi \circ h$, where $h : \mathcal{X} \to \mathbb{R}^c, x \mapsto z$ is the nonlinear function that predicts the logit, and $\phi : \mathbb{R}^c \to [0, 1]^c, z \mapsto \hat{y}^{\text{t}}$ is the mapping function for soft labelling.

We consider two different mapping functions $\phi$:

**Softmax function**. Given the output vector $z_i = h(x_i)$ by the teacher model, we apply softmax activation $\phi_{\text{softmax}}$ to convert the logits into probability vector $\hat{y}_i^{\text{t}}$, and the predicted probability for $j$-th class is computed by comparing the logit $z_{ij}$ with the other logits:

$$\hat{y}_{ij}^{\text{t}} = \phi_{\text{softmax}}(z_i)_j = \frac{\exp(z_{ij}/T)}{\sum_k \exp(z_{ik}/T)}, \tag{2}$$

where $T$ is the temperature that controls the probability distribution over classes.

**Linear function**. Given the logits, $z_{ij}$, computed for each class by teacher model, let $z_{ij'} = \max_l z_{il}$ be the largest logit prediction for $x_i$ and $S_j = \{i | z_{ij} = z_{ij'}\}$, we first use linear function to convert the largest logit $z_{ij'}$ of sample $x_i$ into a probability $r_{ij'}$:

$$r_{ij'} = \frac{z_{ij'} - \min_{i \in S_j} z_{ij'}}{\max_{i \in S_j} z_{ij'} - \min_{i \in S_j} z_{ij'}} + 0.5, \tag{3}$$

where the decision threshold for logits that corresponds to predicted probability $0.5$ is chosen as the smallest logits values of samples whose raw logits predictions are the highest for the $j$-th class. Probability for the other logits can then be linearly distributed as follows:

$$r_{ik} = (1 - r_{ij'})\frac{z_{ik}}{\sum_{k \neq \arg\max_l z_{il}} z_{ik}}, \forall k \neq \arg\max_l z_{il}. \tag{4}$$

## 3.4 Theoretical Analysis

In this subsection, we provide formal analysis on the impact of label smoothing and knowledge distillation on fairness. We first discuss the impact of label smoothing and its connection to reweighing.

For ease of exposition, here we consider binary classification where $y \in \{0, 1\}$. Denote as $y' = \alpha\hat{y}^t + (1 - \alpha)y$ the soft label, the cross entropy loss under soft label can be formulated as

$$L_{soft} = -y'\log(f(x)) - (1 - y')\log(1 - f(x)). \tag{5}$$

And the cross entropy loss under binary label can be formulated as

$$L_{hard} = -y\log(f(x)) - (1 - y)\log(1 - f(x)). \tag{6}$$

Suppose the classifier $f$ makes same predictions on $x$ under both binary and soft labelling at current iteration, we have

$$\begin{aligned}
\Delta = L_{soft} - L_{hard} &= (y - y')\log(f(x)) - (y - y')\log(1 - f(x)) \\
&= \alpha(y - \hat{y}^t)\log\left(\frac{\frac{\exp(z_1)}{\exp(z_0)+\exp(z_1)}}{1 - \frac{\exp(z_1)}{\exp(z_0)+\exp(z_1)}}\right), \\
&= \alpha(y - \hat{y}^t)(z_1 - z_0).
\end{aligned} \tag{7}$$

where $z_i$ is the logit of student network for $i$-th class. Since $\alpha \geq 0$, for $L_{soft} - L_{hard} < 0$, we have $y = 0, z_1 > z_0$ or $y = 1, z_1 < z_0$, which shows that label smoothing down-weighs samples that are incorrectly predicted. For samples that are correctly predicted, larger discrepancy between $y$ and $\hat{y}^t$ indicates larger $\Delta$, which shows that the classifier pays more attention to hard samples that are correctly predicted, and larger $\alpha$ indicates larger discrepancy between $L_{soft}$ and $L_{hard}$.

Similarly, we can compute the induced weight by label smoothing as follows:

$$w(x) = \frac{L_{soft}}{L_{hard}} = y' + (1 - y')\frac{log(1 - f(x))}{log(f(x))}. \tag{8}$$

Here we only consider positive samples for ease of discussion. Let $\tau(x) = \frac{log(1-f(x))}{log(f(x))}$, we have

$$w(x) = (1 - \tau(x))[\alpha\hat{y}^t + (1 - \alpha)y] + \tau(x) \tag{9}$$

Since for $f(x) > 0.5$, we have $\tau(x) > 1$, and $w$ decreases as $\hat{y}^t$ increases. In this way, samples with high soft label but low predicted probability, i.e. positive hard samples that are correctly classified are assigned with higher weight. For samples that are wrongly classified, we have $\tau(x) < 1$, and samples with lower soft label and lower predicted probability, i.e. hard samples that are wrongly classified are assigned with the lowest weight. Besides, since $\hat{y}^t \leq y$, as $\alpha$ increases, $y'$ decreases, and samples that are correctly predicted are assigned with higher weights, while samples that are incorrectly predicted are assigned with lower weights. Similar results also holds for negative samples.

We further derive theoretical analysis to show that our trained with label smoothing and knowledge distillation guarantees fairness in terms of equal odds.

**Theorem 3.1.** *Consider a classifier $f : \mathcal{X} \to [0, 1]$ for binary classification. Denote the classification loss as $L_{soft} = -y'\log(f(x)) - (1 - y')\log(1 - f(x))$ with soft label $y' = \alpha\hat{y}^t + (1 - \alpha)y$, where $\hat{y}^t \in [0, 1]$ is the predicted label from teacher model, $y \in \{0, 1\}$ is the binary label, and $\alpha$ is the balance parameter. The equal odds fairness metrics w.r.t. classifier $f$ is upper bounded by $L_{soft}$.*

# 4 Experiments

## 4.1 Experimental Setup

We validate our model on three benchmark dataset:

- **New Adult**: The Adult reconstruction dataset (Ding et al., 2021) contains 49,531 samples with 14 attributes. The goal is to predict whether an individual's income exceeds certain threshold. During experiments, we set the binary cutoff threshold as $50K$, and we choose race and gender as sensitive attribute.

- **COMPAS**: The COMPAS dataset (Larson et al., 2016) contains 7,215 samples with 11 attributes. Following previous works on fairness (Zafar et al., 2017), we only select black and white defendants in COMPAS dataset, and the modified dataset contains 6,150 samples. The goal is to predict whether a defendant reoffends within two years, and we choose sex and race as sensitive attributes.

- **CelebA**: The CelebA dataset (Liu et al., 2015) contains 202,599 face images, each of resolution $178 \times 218$, with 40 binary attributes. We consider two different binary classifcation tasks on CelebA dataset: predicting attractiveness with gender as sensitive attribute, and predicting gender with age as sensitive attribute.

We implement our method in PyTorch 1.10.1 with one NVIDIA RTX-3090 GPU. We use accuracy as evaluation metric, and we apply disparate impact (Kamiran and Calders, 2012) and equalized odds (Hardt et al., 2016) as fairness metrics. We build the teacher model using ResNet-152 (He et al., 2016) and student model using ResNet-18 (He et al., 2016). We compare our method with the following related methods:

- **Teacher model**: Teacher network trained on binary label. The teacher model is expected to overfit training data.

- **Student model (binary)**: Student network trained on binary label.

- **Student model (softmax)**: Student network trained on soft label obtained by the output of teacher model with softmax activation as in (2).

- **Student model (linear)**: Student network trained on soft label obtained by the output of teacher model with linear normalization as in (3).

- **Distributionally robust optimization (DRO)**: Student network with distributionally robust optimization (Hashimoto et al., 2018). The network is trained on binary label.

- **Adversarially reweighted learning (ARL)**: Student network with adversarially reweighted learning (Lahoti et al., 2020). The network is trained on binary label. The adversarial network is chosen as one linear layer as suggested by the authors (Lahoti et al., 2020).

- **FairRF**: Student network with FairRF (Zhao et al., 2022). The network is trained on binary label. We include results of FairRF only on new Adult and COMPAS dataset, as it is hard to regulate sensitive-related features for images.

We repeat experiments on each dataset five times and report the average results. To avoid large discrepancies in testing data, before each repetition, we randomly spilt data into 50% training data, 10% validation data and 40% test data. All the methods evaluated are trained and tested on the same data partitions each time. For student model trained on softmax label, the temperature is tuned to find the best validation accuracy. The hyperparameters of comparing methods are tuned with binary search to find global minimum, as suggested in the original paper (Hashimoto et al., 2018).

## 4.2 Experimental Results

Experimental results are shown in Table 2-7. On all three datasets, both DRO and ARL show a drop in accuracy, especially for DRO which strictly follows the formulation of Max-Min fairness. However, both methods do not show notable improvement in fairness. This is in line with our hypothesis that Max-Min fairness could be too strict to achieve expected improvement in group fairness. In comparison, our method does not induce as much decrease in accuracy, but achieves significant improvement in terms of fairness on all three datasets, especially for equalized odds. This shows the

| Method | Accuracy | Disparate impact | Equalized odds |
|---|---|---|---|
| Teacher | 84.41% | 20.27% | 39.64% |
| Student (with hard label) | 64.13±0.32% | 23.27±2.43% | 38.34±3.37% |
| DRO (Hashimoto et al., 2018) | 62.67±0.73% | 21.41±2.19% | 30.43±3.24% |
| ARL (Lahoti et al., 2020) | 63.23±0.47% | 21.37±3.46% | 29.46±1.74% |
| FairRF (Zhao et al., 2022) | 63.26±0.83% | 21.47±1.76% | 25.67±2.63% |
| Student (with softmax label) | 63.47±0.44% | 19.52±2.46% | 21.32±1.97% |
| Student (with linear label) | 63.34±0.46% | 20.27±2.34% | 20.31±2.62% |

Table 2: Results on COMPAS dataset with sensitive attribute *race*.

| Method | Accuracy | Disparate impact | Equalized odds |
|---|---|---|---|
| Teacher | 84.41% | 19.42% | 34.41% |
| Student (with hard label) | 64.13±0.32% | 19.17±2.33% | 20.25±2.53% |
| DRO (Hashimoto et al., 2018) | 62.67±0.73% | 19.62±2.27% | 18.75±2.18% |
| ARL (Lahoti et al., 2020) | 63.23±0.47% | 18.87±3.32% | 19.14±2.56% |
| FairRF (Zhao et al., 2022) | 63.26±0.83% | 17.23±1.84% | 18.74±2.21% |
| Student (with softmax label) | 63.37±0.44% | 16.63±1.67% | 14.32±2.47% |
| Student (with linear label) | 63.34±0.46% | 16.14±1.83% | 15.13±2.34% |

Table 3: Results on COMPAS dataset with sensitive attribute *sex*.

| Method | Accuracy | Disparate impact | Equalized odds |
|---|---|---|---|
| Teacher | 90.63% | 14.43% | 17.76% |
| Student (with hard label) | 85.42±0.46% | 12.31±1.46% | 14.43±1.24% |
| DRO (Hashimoto et al., 2018) | 82.64±0.62% | 12.27±2.27% | 13.69±1.34% |
| ARL (Lahoti et al., 2020) | 83.37±0.46% | 12.34±1.76% | 14.02±1.43% |
| FairRF (Zhao et al., 2022) | 83.74±0.86% | 11.37±1.46% | 11.23±1.42% |
| Student (with softmax label) | 84.63±0.47% | 10.63±1.34% | 10.34±1.22% |
| Student (with linear label) | 84.27±0.31% | 10.21±1.52% | 10.57±1.64% |

Table 4: Results on new Adult dataset with sensitive attribute *race*.

| Method | Accuracy | Disparate impact | Equalized odds |
|---|---|---|---|
| Teacher | 90.63% | 19.42% | 24.47% |
| Student (with hard label) | 85.42±0.32% | 17.63±1.49% | 18.82±1.82% |
| DRO (Hashimoto et al., 2018) | 82.64±0.83% | 17.21±1.63% | 17.63±2.27% |
| ARL (Lahoti et al., 2020) | 83.37±0.42% | 16.62±1.47% | 16.52±1.96% |
| FairRF (Zhao et al., 2022) | 83.74±0.86% | 16.37±2.41% | 13.54±1.26% |
| Student (with softmax label) | 84.63±0.43% | 15.63±1.46% | 12.23±1.83% |
| Student (with linear label) | 84.27±0.64% | 15.56±1.54% | 11.59±1.74% |

Table 5: Results on new Adult dataset with sensitive attribute *gender*.

| Method | Accuracy | Disparate impact | Equalized odds |
|---|---|---|---|
| Teacher | 86.67% | 18.46% | 31.47% |
| Student (with hard label) | 81.67±0.63% | 18.62±1.34% | 26.42±1.61% |
| DRO (Hashimoto et al., 2018) | 76.62±0.26% | 17.26±1.82% | 23.26±2.68% |
| ARL (Lahoti et al., 2020) | 77.14±0.75% | 17.23±2.48% | 21.41±2.21% |
| Student (with softmax label) | 80.87±0.14% | 15.27±1.71% | 11.43±1.25% |
| Student (with linear label) | 80.76±0.73% | 14.47±1.64% | 10.62±1.10% |

Table 6: Results of attractiveness classification on CelebA dataset with sensitive attribute *gender*.

effectiveness of knowledge from teacher model and error-based reweighing for soft labelling. We observe that the pre-trained overfitted teacher model achieves higher accuracy on all three datasets, which is in line with the expectation of knowledge distillation. However, the overfitted teacher model does not improve fairness, and the equalized odds of teacher model for COMPAS and new Adult dataset are much worse than baseline student model, and we conclude that improvement in fairness with knowledge distillation is due to soft labelling, instead of complexity of teacher model.

| Method | Accuracy | Disparate impact | Equalized odds |
|---|---|---|---|
| Teacher | 94.42% | 17.31% | 17.46% |
| Student (with hard label) | 90.43±0.23% | 16.67±2.26% | 17.32±1.78% |
| DRO (Hashimoto et al., 2018) | 74.46±0.37% | 16.73±3.15% | 16.45±2.26% |
| ARL (Lahoti et al., 2020) | 84.27±0.94% | 16.21±1.86% | 15.54±1.57% |
| Student (with softmax label) | 89.21±0.15% | 12.23±1.84% | 10.13±2.14% |
| Student (with linear label) | 89.42±0.68% | 13.16±2.21% | 9.64±2.14% |

Table 7: Results of gender classification on CelebA dataset with sensitive attribute *age*.

## 4.3 Parameter Analysis

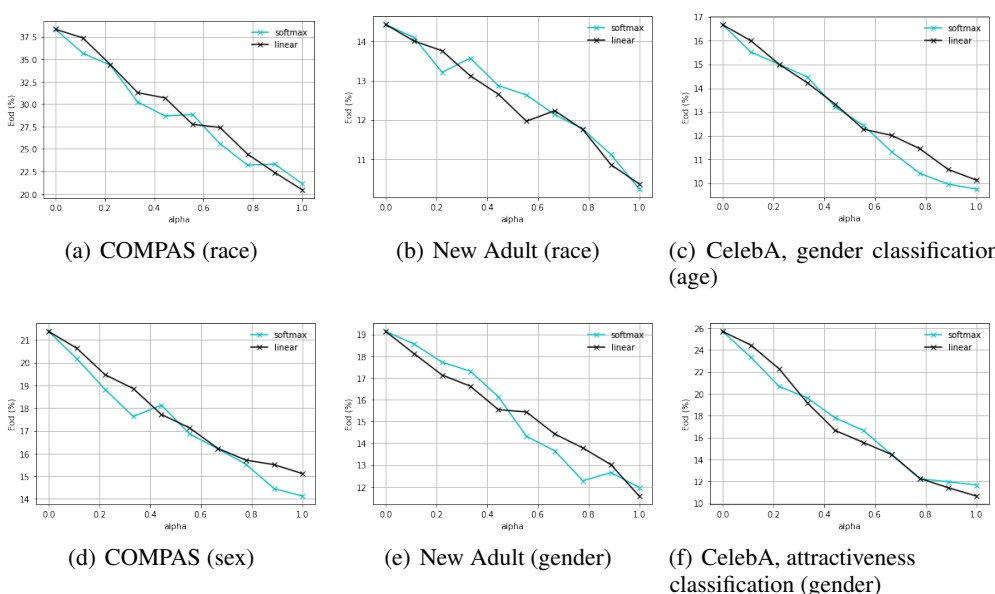

(a) COMPAS (race)

(b) New Adult (race)

(c) CelebA, gender classification (age)

(d) COMPAS (sex)

(e) New Adult (gender)

(f) CelebA, attractiveness classification (gender)

Figure 2: Change of equalized odds as $\alpha$ varies on COMPAS, new adult and CelebA dataset. The sensitive attributes for each setting are shown in parenthesis.

We move on to discuss the influence of the trade-off parameter $\alpha$ in (1) on accuracy and fairness. It is straightforward to see that as $\alpha$ varies, the corresponding loss term can be treated as linear interpolation between binary cross-entropy loss and soft cross-entropy loss, and increasing $\alpha$ increase the role of soft label in knowledge distillation, which we observe to help improve fairness as in Fig. 2. As shown in Fig. 3, as $\alpha$ increases, there is a small decrease in accuracy, and on all three datasets, as $\alpha$ increases, under both softmax and linear labelling, although with fluctuation under certain variation of $\alpha$ on COMPAS and new Adult datasets, overall the equalized odds regarding different sensitive attributes show downtrend, which validates the effectiveness of soft labelling.

## 5 Conclusion

Fairness without demographics has been an important yet less studied task. In this paper, we propose to solve this problem through knowledge distillation. We consider a pre-trained complex teacher model that is overfitted to training data, and we use the normalized logits as soft label to train a simple student model. We consider two different normalization functions, the softmax function and linear function. We theoretically discuss the connection between soft labelling and reweighing, and we prove theoretically that with knowledge distillation, the training objective of student model upper-bounds several fairness metrics. We validate from preliminary experiments on new Adult dataset the effectiveness of soft labelling in improving fairness, and we show from experiments that knowledge distillation achieves remarkable improvement in fairness with relatively small sacrifice in accuracy.

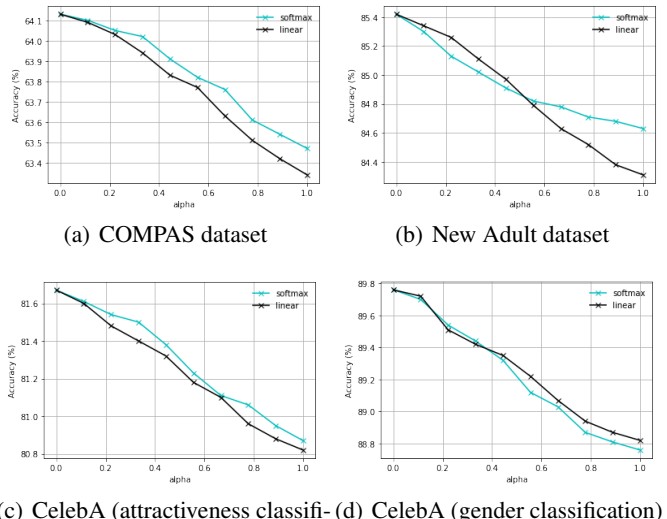

| (a) COMPAS dataset | (b) New Adult dataset |
| (c) CelebA (attractiveness classifi-cation) | (d) CelebA (gender classification) |

Figure 3: Change of accuracy as $\alpha$ varies on COMPAS, new Adult and CelebA dataset.

There are two main directions of interest for future work. First is the alternatives of mapping function: can we define other mapping functions for the logits such that the student model achieve better fairness? Understanding the influence of mapping functions is crucial in extending our work. Another topic is robustness of our method to distributional shift, especially the robustness of fairness, as robustness to distributional shift has been an important but challenging problem in the context fairness. Other topics include few-shot tweaking with partially available sensitive attributes.

## Acknowledgements

This work was partially supported by the EMBRIO Institute, contract #2120200, a National Science Foundation (NSF) Biology Integration Institute, Purdue's Elmore ECE Emerging Frontiers Center, and NSF IIS #1955890, IIS #2146091.

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
