# A    Proof of Theorem 3.1

*Proof.* Consider a given data $(x, y)$, where $x \in \mathcal{X}$ is the input data, and $y \in \{0, 1\}$ is the binary label. Denote $y^t \in [0, 1]$ as the predicted output from the teacher model, and $f : \mathcal{X} \to [0, 1], x \mapsto \hat{y}^s$ as the student model, then the mixup loss for the data $(x, y)$ can be formulated as:

$$L_{soft}(x, y) = -(\alpha y^t + (1 - \alpha)y) \log(f(x)) - (1 - (\alpha y^t + (1 - \alpha)y)) \log(1 - f(x)),$$

where $\alpha$ is the balance parameter.

Here we consider binary sensitive group, and denote $\mathcal{X}_a$ as the data distribution from the sensitive group $A = a, a \in \{0, 1\}$. Then the training objective of the student model $f$ is:

$$J_{opt} = \min_f \left[ \mathbb{E}_{x \sim \mathcal{X}_0}[L_{soft}(x, y)] + \mathbb{E}_{x \sim \mathcal{X}_1}[L_{soft}(x, y)] \right]$$

$$\implies \min_f \left[ \alpha \mathbb{E}_{x \sim \mathcal{X}_0}[L_{BCE}(x, y)] + \alpha \mathbb{E}_{x \sim \mathcal{X}_1}[L_{BCE}(x, y)] \right.$$

$$\left. + (1 - \alpha)\mathbb{E}_{x \sim \mathcal{X}_0}[L_{BCE}(x, y^t)] + (1 - \alpha)\mathbb{E}_{x \sim \mathcal{X}_1}[L_{BCE}(x, y^t)] \right],$$

where $L_{BCE}(x, y) = -y \log(f(x)) - (1 - y) \log(1 - f(x))$ is the binary cross-entropy loss.

As proved by Feng et al. (2021), the binary cross-entropy loss $L_{BCE}(x, y)$ is lower-bounded by its corresponding mean absolute error $L_{MAE}(x, y)$, i.e., $L_{BCE}(x, y) \geq \frac{1}{2}L_{MAE}(x, y) = \frac{1}{2}|x - y|$.

Furthermore, consider the classification loss

$$L_{cls}(x, y) = \mathbb{1}[(f(x) - 0.5)(y - 0.5) < 0],$$

when the data is correctly classified, we have $L_{cls}(x, y) = 0$ and $L_{MAE}(x, y) \geq L_{cls}(x, y)$; while when the data is wrongly classified, we have $L_{cls}(x, y) = 1$ and $L_{MAE}(x, y) \geq L_{cls}(x, y) - 0.5$. According to Ji and Zhu (2020), $min(y, y^t) < \hat{y}^s < max(y, y^t)$, thus $L_{MAE}(x, y^t) \geq L_{cls}(x, y) - 0.5$ also holds true.

Thus we have:

$$J_{opt} \geq \frac{\alpha}{2}\mathbb{E}_{x \sim \mathcal{X}_0}[|x - y|] + \frac{\alpha}{2}\mathbb{E}_{x \sim \mathcal{X}_1}[|x - y|] + \frac{1 - \alpha}{2}\mathbb{E}_{x \sim \mathcal{X}_0}[|x - y^t|] + \frac{1 - \alpha}{2}\mathbb{E}_{x \sim \mathcal{X}_1}[|x - y^t|]$$

$$\geq \frac{\alpha}{2}\mathbb{E}_{x \sim \mathcal{X}_0}[L_{cls}(x, y)] + \frac{\alpha}{2}\mathbb{E}_{x \sim \mathcal{X}_1}[L_{cls}(x, y)]$$

$$+ \frac{1 - \alpha}{2}\mathbb{E}_{x \sim \mathcal{X}_0}[L_{cls}(x, y^t)] + \frac{1 - \alpha}{2}\mathbb{E}_{x \sim \mathcal{X}_1}[L_{cls}(x, y^t)]$$

$$\geq \mathbb{E}_{x \sim \mathcal{X}_0}[L_{cls}(x, y)] + \mathbb{E}_{x \sim \mathcal{X}_1}[L_{cls}(x, y)] - 1$$

$$= r_{pos}\text{FPR}_0 + r_{neg}\text{FNR}_0 + r_{pos}\text{FPR}_1 + r_{neg}\text{FNR}_1 - 1$$

$$\geq r_{pos}|\text{FPR}_0 - \text{FPR}_1| + r_{neg}|\text{FNR}_0 - \text{FNR}_1|,$$

where $FPR_a$ and $FNR_a$ are the false positive rate and false negative rate of the sensitive group $A = a, a \in \{0, 1\}$.

$\square$

# B    Ablation Study

We include more results on teacher model and teacher model + {DRO (Hashimoto et al., 2018) /ARL (Lahoti et al., 2020) / FairRF (Zhao et al., 2022) /our knowledge distillation} in Tab. 8-13. Effect of our label smoothing can be observed by comparing between "Teacher (with hard label)" and "Teacher (with softmax/linear label)" in the 6 tables. Here the capacity is the same, the only difference is the label smoothing. compared with student model trained by the same method, the teacher model achieves better accuracy and comparable fairness. Here the training method is the same, only difference is capacity. This shows that increasing model capacity helps in improving accuracy, but does not have much influence on fairness.

| Method | Accuracy | Disparate impact | Equalized odds |
|---|---|---|---|
| Teacher (with hard label) | 64.85±0.93% | 23.41±2.36% | 37.34±2.85% |
| Teacher+DRO | 63.12±0.61% | 21.36±2.25% | 30.43±2.94% |
| Teacher+ARL | 63.74±0.65% | 21.42±2.85% | 29.45±1.83% |
| Teacher+FairRF | 63.69±0.47% | 21.26±1.76% | 25.19±2.47% |
| Teacher (self-distillation with softmax label) | 64.12±0.63% | 19.47±2.25% | 20.85±2.61% |
| Teacher (self-distillation with linear label) | 63.97±0.62% | 21.24±2.62% | 20.16±2.34% |

Table 8: Results on COMPAS dataset with sensitive attribute *race*. Here the Teacher model used in the first 4 rows, and the teacher model in row 5-6 after self-distillation are not trained to overfit to the data.

| Method | Accuracy | Disparate impact | Equalized odds |
|---|---|---|---|
| Teacher (with hard label) | 64.85±0.93% | 19.35±2.24% | 20.73±2.16% |
| Teacher+DRO | 63.12±0.61% | 19.57±2.15% | 18.69±2.27% |
| Teacher+ARL | 63.74±0.65% | 18.64±3.21% | 19.15±2.19% |
| Teacher+FairRF | 63.69±0.47% | 17.21±1.58% | 18.16±2.21% |
| Teacher (self-distillation with softmax label) | 64.12±0.63% | 16.42±1.72% | 13.74±2.61% |
| Teacher (self-distillation with linear label) | 63.97±0.62% | 16.32±1.51% | 14.94±2.42% |

Table 9: Results on COMPAS dataset with sensitive attribute *sex*. The classifier is chosen as teacher network.

| Method | Accuracy | Disparate impact | Equalized odds |
|---|---|---|---|
| Teacher (with hard label) | 85.17±0.92% | 12.46±1.57% | 14.65±1.62% |
| Teacher+DRO | 83.64±0.73% | 12.34±2.23% | 13.82±1.51% |
| Teacher+ARL | 83.67±0.73% | 11.84±1.56% | 14.23±1.42% |
| Teacher+FairRF | 84.24±0.67% | 11.14±1.56% | 10.87±1.45% |
| Teacher (self-distillation with softmax label) | 84.47±0.94% | 10.49±1.53% | 10.21±1.67% |
| Teacher (self-distillation with linear label) | 84.12±0.65% | 10.43±1.58% | 10.64±1.69% |

Table 10: Results on new Adult dataset with sensitive attribute *race*. The classifier is chosen as teacher network.

| Method | Accuracy | Disparate impact | Equalized odds |
|---|---|---|---|
| Teacher (with hard label) | 85.17±0.92% | 17.59±1.35% | 19.24±2.23% |
| Teacher+DRO | 83.64±0.73% | 16.67±1.54% | 17.63±2.16% |
| Teacher+ARL | 83.67±0.73% | 16.35±1.47% | 16.21±1.73% |
| Teacher+FairRF | 84.24±0.67% | 16.67±1.49% | 13.11±1.21% |
| Teacher (self-distillation with softmax label) | 84.47±0.94% | 15.34±1.61% | 12.68±2.31% |
| Teacher (self-distillation with linear label) | 84.12±0.65% | 15.62±2.13% | 11.87±1.68% |

Table 11: Results on new Adult dataset with sensitive attribute *gender*. The classifier is chosen as teacher network.

| Method | Accuracy | Disparate impact | Equalized odds |
|---|---|---|---|
| Teacher (with hard label) | 84.65±1.84% | 18.87±2.52% | 27.36±2.37% |
| Teacher+DRO | 78.14±1.27% | 17.36±1.52% | 23.24±2.46% |
| Teacher+ARL | 79.61±0.93% | 17.41±1.86% | 21.26±2.14% |
| Teacher (self-distillation with softmax label) | 82.69±1.61% | 15.24±1.83% | 11.65±1.85% |
| Teacher (self-distillation with linear label) | 83.15±1.48% | 15.34±1.57% | 10.74±2.42% |

Table 12: Results of attractiveness classification on CelebA dataset with sensitive attribute *gender*. The classifier is chosen as teacher network.

| Method | Accuracy | Disparate impact | Equalized odds |
|---|---|---|---|
| Teacher (with hard label) | 91.46±2.15% | 16.92±1.51% | 17.45±2.33% |
| Teacher+DRO | 79.26±1.25% | 16.51±2.68% | 16.27±2.15% |
| Teacher+ARL | 86.34±0.82% | 16.43±1.24% | 15.52±1.69% |
| Teacher (self-distillation with softmax label) | 90.47±2.16% | 12.16±1.67% | 10.25±1.83% |
| Teacher (self-distillation with linear label) | 90.43±2.38% | 12.84±1.49% | 10.17±2.25% |

Table 13: Results of gender classification on CelebA dataset with sensitive attribute *age*. The classifier is chosen as teacher network.