# OpenReview forum: "Fairness without Demographics through Knowledge Distillation"
_NeurIPS.cc/2022/Conference — NeurIPS 2022 Accept_

### Official Review · Reviewer_Sqvi · 2022-07-08

**Rating:** 6
**Confidence:** 4
**Soundness:** 3 good
**Presentation:** 4 excellent
**Contribution:** 3 good

**Summary:**

This paper proposes to use knowledge distillation to improve fairness in the student model. The key idea is to use soft labels from the teacher, which essentially poses a smoothness constraint in the student model, to improve fairness (disparate impact and equalized odds).

The authors did experiments on 3 datasets, new Adult, COMPAS, and CelebA, and showed that the proposed method improves upon existing baselines, including a DRO based approach and an adversarial re-weighting based approach.

**Questions:**

- Can the authors include a baseline that does direct label smoothing on hard labels in training?

- Can the authors add capacity-separated baselines, like Teacher + DRO / ARL?

- Would a combination of Teacher + DRO / ARL + label smoothing further improve fairness? i.e., you distill a model from Teacher + DRO/ARL through the label smoothing you proposed.

- Minor: in Lahoti et al., 2020, the AUC doesn't seem to decrease significantly across Adult (the older dataset) and COMPAS, why it shows a significant drop on the student model compared to student (with hard label) across Table 2/3/4/5?

**Limitations:**

Discussed in Section 5.

**Strengths And Weaknesses:**

Strengths:
- This paper is well written. The key idea is clearly presented. The authors first show that label smoothing can help fairness, and then proposed a reasonable approach to practically implement this through knowledge distillation when continuous labels are not directly available.
- The idea of imposing smoothness in the model to improve fairness through knowledge distillation is novel to the best of my knowledge.
- The authors did experiments on 3 datasets, and the results show that the proposed method works universally well across all datasets, outperforming existing DRO and adversarial-reweighting baselines.

Weaknesses:
- Since the authors study label smoothing, one obvious baseline seems to be missing, which is training the model by smoothing the binary labels directly from [1], i.e., using $y(1-a) + a/K$ where $a$ controls the smoothness and $K$ is the number of classes.

[1] When Does Label Smoothing Help? Muller et al. NeurIPS 2019.

- By looking at the results, there might actually be multiple components that contribute to the final fairness which are entangled together. (1) label smoothing does help the final fairness, as mentioned in Section 3.2. (2) model capacity also seems to contribute to the fairness, as the student trained with hard label already has much better fairness across multiple datasets.

For completeness, I think the authors should disentangle those factors, i.e., separating the tables into two parts,
- first part is high-capacity model only (i.e., the teacher), and there should be baselines like a) Teacher (with hard label), b) Teacher with direct label smoothing, c) Teacher with DRO/ARL;
- then the second part should be low-capacity model only (i.e., the student), and you should list all the results currently presented.

This should give people a clearer picture about the contribution of each component.

---

> ### Author Response · Authors · 2022-08-02
> **Response to reviewer Sqvi**
>
> Thanks for the comment.
>
> * **More experimental results:** We include results on teacher model and teacher model + {label smoothing[1] /DRO[2] /ARL[3] /our label smoothing} in the 6 tables below (Table 1-4 can be found [here](https://docs.google.com/document/d/1oQQhn9hqrD-th0W1ygGpmyA58BfYkuzvqfsyY_ukxv4/edit?usp=sharing)). "Teacher" refers to teacher model structure, and "student" refers to the student model structure. Experiments are repeated 5 times for each method.
>
> * **Contribution of each component:** (1) effect of our label smoothing can be observed by comparing between "Student (with hard label)" and "Student (with softmax/linear label)" in the 6 tables below. Here the capacity is the same, the only difference is the label smoothing. Compared with baseline label smoothing method, our method achieves comparable performance in terms of accuracy and better improvement in fairness under both teacher and student model capacities, which shows the effectiveness of our label smoothing method. (2) effect of model capacity: compared with student model trained by the same method, the teacher model achieves better accuracy and comparable fairness. Here the training method is the same, only difference is capacity. This shows that increasing model capacity helps in improving accuracy, but does not have much influence on fairness.
>
> **Table 5: CelebA (sensitive attribute: _gender_)**
> Method|Accuracy(\%)|Disparate impact(\%)|Equalized odds(\%)
> -|-|-|-
> Teacher (with hard label)|84.65$\pm$1.84|18.87$\pm$2.52| 27.36$\pm$2.37
> Teacher+DRO [2] | 78.14$\pm$1.27| 17.36$\pm$1.52|23.24$\pm$2.46
> Teacher+ARL [3] |79.61$\pm$0.93|17.41$\pm$1.86| 21.26$\pm$2.14
> Teacher+Label smoothing by [1] |82.69$\pm$1.61| 18.47$\pm$1.69|25.47$\pm$1.84
> Teacher (self-distillation with softmax label by Eq. 2)|84.47$\pm$0.94| 15.34$\pm$1.61| 11.65$\pm$1.85
> Teacher (self-distillation with linear label by Eq. 3)|83.15$\pm$1.48| 15.34$\pm$1.57|10.74$\pm$2.42
> Student (with hard label) |81.67$\pm$0.63| 18.62$\pm$1.34|26.42$\pm$1.61
> Student+DRO [2] | 76.62$\pm$0.26|17.26$\pm$1.82|23.26$\pm$2.68
> Student+ARL [3] |77.14$\pm$0.75|17.23$\pm$2.48| 21.41$\pm$2.21
> Student (with softmax label by Eq. 2)  | 80.87$\pm$0.14|15.27$\pm$1.71|11.43$\pm$1.25
> Student (with linear label by Eq. 3)  |80.76$\pm$0.73|14.47$\pm$1.64|10.62$\pm$1.10
>
> **Table 6: CelebA (sensitive attribute: _age_)**
> Method|Accuracy(\%)|Disparate impact(\%)|Equalized odds(\%)
> -|-|-|-
> Teacher (with hard label)|91.46$\pm$2.15|16.92$\pm$1.51| 17.45$\pm$2.33
> Teacher+DRO [2] |79.26$\pm$1.25| 16.51$\pm$2.68|16.27$\pm$2.15
> Teacher+ARL [3] |86.34$\pm$0.82|16.43$\pm$1.24| 15.52$\pm$1.69
> Teacher+Label smoothing by [1] |88.62$\pm$1.46| 17.25$\pm$1.13|16.52$\pm$2.36
> Teacher (self-distillation with softmax label by Eq. 2)|90.47$\pm$2.16|12.16$\pm$1.67| 10.25$\pm$1.83
> Teacher (self-distillation with linear label by Eq. 3)|90.43$\pm$2.38| 12.84$\pm$1.49|10.17$\pm$2.25
> Student (with hard label) |90.43$\pm$0.23| 16.67$\pm$2.26|17.32$\pm$1.78
> Student+DRO [2] | 74.46$\pm$0.37|16.73$\pm$3.15|16.45$\pm$2.26
> Student+ARL [3] |84.27$\pm$0.94|16.21$\pm$1.86| 15.54$\pm$1.57
> Student (with softmax label by Eq. 2)  | 89.21$\pm$0.15|12.23$\pm$1.84|10.13$\pm$2.14
> Student (with linear label by Eq. 3)  |89.42$\pm$0.68|13.16$\pm$2.21|9.64$\pm$2.14
>
> * **Combination of Teacher + DRO [2]/ARL [3] + our label smoothing:** It is possible to achieve better fairness with our label smoothing + DRO [2]/ARL [3]. In the table below we take ARL as an example to validate this hypothesis. The results show that the soft label from a fair teacher model may help with the fairness of student model. Thanks for the suggestion and will further study this in future work.
>
> Dataset|Sensitive attribute|Accuracy(\%)|Disparate impact(\%)|Equalized odds(\%)
> -|-|-|-|-
> COMPAS |race | 64.45$\pm$0.48|23.16$\pm$2.42|18.84$\pm$2.65
> COMPAS |sex|64.45$\pm$0.48|16.27$\pm$1.46|14.53$\pm$2.41
> New Adult|race|84.59$\pm$0.47|9.65$\pm$1.48|9.45$\pm$1.83
> New Adult|gender|84.59$\pm$0.47|13.82$\pm$2.26|10.34$\pm$1.73
> CelebA|gender|81.86$\pm$1.28|13.41$\pm$1.17|9.19$\pm$1.45
> CelebA|age|89.21$\pm$1.67|13.17$\pm$2.25|9.31$\pm$1.64
>
> * **Minor:** The metric used in [3] is AUC instead of accuracy. The accuracy drop in Table 2/3/4/5 of our main paper is around $1\%-2\%$ for DRO and ARL compared with baseline student network. This could be due to the difference in network structure, dataset selection (older Adult vs new Adult), etc.
>
> **[Reference]**
>
> [1] Müller, R., Kornblith, S., \& Hinton, G. E. When does label smoothing help?. In NeurIPS 2019.
>
> [2] Hashimoto, T., Srivastava, M., Namkoong, H., & Liang, P. Fairness without demographics in repeated loss minimization. In ICML 2018.
>
> [3] Lahoti, P., Beutel, A., Chen, J., Lee, K., Prost, F., Thain, N., ... & Chi, E. Fairness without demographics through adversarially reweighted learning. In NeurIPS 2020.

---

> > ### Comment · Reviewer_Sqvi · 2022-08-09
> > **Thanks for the response**
> >
> > Thanks for adding those additional baselines and I think all my concerns are well addressed.

---

> > > ### Author Response · Authors · 2022-08-09
> > > **Thank You**
> > >
> > > Dear Reviewer,
> > >
> > > Thank you for taking the time and effort to review our work. We sincerely appreciate your constructive feedback and recognition of our work!
> > >
> > > Best,
> > > Authors

---

### Official Review · Reviewer_V6Vx · 2022-07-09

**Rating:** 4
**Confidence:** 4
**Soundness:** 2 fair
**Presentation:** 3 good
**Contribution:** 2 fair

**Summary:**

This paper introduces a new method to achieve considerable improvement in group fairness through knowledge distillation. The solution is to use the soft labels from an overfitted teacher model to train a student model, and the paper analyzes the fairness of the solution theoretically. The experiment results on several extensively-used datasets indicate that the proposed method enhances the fairness in the outcome without loss of Accuracy. The main contribution of the paper is that the authors propose an alternative method when demographic information is not available in the training set.

**Questions:**

As for the questions and suggestions, please refer to the Weakness in the “Strengths And Weaknesses”. Nevertheless, I still have some concerns:
1. Are the teacher model and student model both trained without the demographic data? Why the teacher model is expected to overfit training data (in line 250)?
2. Did the paper miss some important baselines or references? For example,
(1)	Celis, L. E., Huang, L., Keswani, V., & Vishnoi, N. K. (2021, July). Fair classification with noisy protected attributes: A framework with provable guarantees. In International Conference on Machine Learning (pp. 1349-1361). PMLR.
(2)	Celis, L. E., Mehrotra, A., & Vishnoi, N. (2021). Fair classification with adversarial perturbations. Advances in Neural Information Processing Systems, 34, 8158-8171.
(3)	Tianxiang Zhao, Enyan Dai, Kai Shu, and Suhang Wang. 2022. Towards Fair Classifiers Without Sensitive Attributes: Exploring Biases in Related Features. In Proceedings of the Fifteenth ACM International Conference on Web Search and Data Mining (WSDM '22). Association for Computing Machinery, New York, NY, USA, 1433–1442.
Other details are as above.


**Limitations:**

Please refer to the main reviews.

**Strengths And Weaknesses:**

Strengths:
1. The paper is clearly organized and well written.
2. The authors propose an alternative method when demographic information is not available in the training set. The paper analysis the impact of label smoothing and knowledge distillation on fairness theoretically.
3. The experiments demonstrate the effectiveness of the proposed framework, i.e. the improvement of fairness without loss of Accuracy performance.
4. The research direction is interesting and the proposed method is relevant to many domains.


Weaknesses：
1. At the beginning of the paper, it needs a better description of the motivation and novelty; it's hard to know where the challenge is to solve the fairness problem through knowledge distillation.
2. Besides, except for the theoretical proof of the knowledge distillation on the fairness problem, the paper needs to introduce the reason why it is useful empirically or intuitively to help readers understand better.
3. The interpretability of the proposed method should be discussed. For example, when demographic information is not available, how to discriminate different groups, such as gender.
4. The procedure of the proposed method by authors on the fairness is not shown logically and intuitively. Adding a figure and refining relevant paragraphs to describe the procedure of how the knowledge distillation can improve fairness may help readers who is not familiar with fairness understand better.
5. There are some typos in this paper. For example, the “individual” shall be “individual” in line 148, and the formula in line 227 may be wrong.

---

> ### Author Response · Authors · 2022-08-02
> **Response to reviewer V6Vx**
>
> Thanks. We'll fix the typo and include the suggested reference in final version.
>
> * **Motivation and novelty:** The motivation of our method is to improve fairness without accessing sensitive information. Much of current literature on fairness without demographics formulates the problem as Max-Min fairness objectives; however, this formulation could be too strict to achieve expected improvement in group fairness. Instead, we seek to consider the problem from a new perspective (i.e., label smoothing), and in this way we ensure hard samples in training set are properly addressed to improve fairness.
>
> * **Empirical or intuitive introduction:** We use preliminary results in Section 3.2 to show the effectiveness of label smoothing. From Table 1 of our main paper, both linear and softmax labelling improves fairness, which shows the feasibility of label smoothing for fairness without demographics. In real applications, since ground truth soft labels are not applicable for most datasets, we instead consider using knowledge distillation to obtain soft labels and train the student model with soft labels by teacher model.
>
> * **Interpretability of our method:** We first clarify that even when the sensitive attribute is not available as direct input, there still can be fairness concern in the prediction. Due to distributional disparities between different sensitive groups, machine learning models can easily make biased decisions against certain sensitive groups even through no sensitive information is used during training [1,2]. Therefore, it's necessary to introduce extra constraint to ensure fairness. Second, in our problem setting, it is hard to find an accurate estimation of sensitive information when only label information is available. One of our objectives for fairness without demographics is to improve fairness under different sensitive attributes, and due to distributional disparities between different sensitive groups, it's not easy to achieve fairness for different sensitive groups simultaneously under estimation of certain sensitive attribute. Instead, we address this problem by label smoothing, where our assumption is that focusing on hard samples in training set will benefit disadvantaged group(s) (similar to the ideas of DRO [3] and ARL [4]), and in this way we reduce disparities between different sensitive groups. During testing, the sensitive information is needed to compute fairness metrics (disparate impact, equalized odds, etc.)
>
> * **Figure description:** Thanks! We show a [demonstration figure](https://docs.google.com/document/d/1PnnOy6fpVCkvi0DekK1XoHwWb2BA_Fsee5kuGuKyDiQ/edit?usp=sharing) of our knowledge distillation method. We'll include the figure in final paper.
>
> * **Q1:** Both student and teacher models are trained **without** sensitive information. It's a common strategy to make teacher model overfitted to training set s.t. the learned knowledge better transfers to the student model [5,6,7].
>
> * **Q2:** Thanks. We 'll add the suggested papers [Celis et al., ICML 2021] and [Celis et al., NeurIPS 2021] in related work. [Celis et al., ICML 2021] studies fair classification under noisy sensitive attribute, and [Celis et al., NeurIPS 2021] studies fair classification under adversarially perturbed sensitive attribute. As both methods still require access to sensitive attribute, we don't include the two for experimental comparison. Part of the most recent and most related method [Zhao et al., WSDM 2022] (as suggested by the reviewer) are shown below. Experiments are conducted under same setting as our method. We'll include full results in final paper.
>
> Dataset|Method|Sensitive attribute|Accuracy(\%)|Disparate impact(\%)|Equalized odds(\%)
> -|-|-|-|-|-
> COMPAS|FairRF|race|63.26$\pm$0.83|21.47$\pm$1.76|25.67$\pm$2.63
> COMPAS|Ours|race|63.34$\pm$0.46|20.27$\pm$2.34|20.31$\pm$2.62
> COMPAS|FairRF|sex| 63.26$\pm$0.83|17.23$\pm$1.84|18.74$\pm$2.21
> COMPAS|Ours|sex| 63.34$\pm$0.46|16.14$\pm$1.83|15.13$\pm$2.34
> New Adult|FairRF|race| 83.74$\pm$0.86|11.37$\pm$1.46|11.23$\pm$1.42
> New Adult|Ours|race|84.27$\pm$0.31|10.21$\pm$1.52|10.57$\pm$1.64
> New Adult|FairRF|gender|83.74$\pm$0.86|16.37$\pm$2.41|13.54$\pm$1.26
> New Adult|Ours|gender|84.27$\pm$0.64|15.56$\pm$1.54|11.59$\pm$1.74
>
> **[Reference]**
>
> [1] Corbett-Davies, S. et al. The measure and mismeasure of fairness: A critical review of fair machine learning. arXiv preprint arXiv:1808.00023. 2018.
>
> [2] Hajian, S. et al. A methodology for direct and indirect discrimination prevention in data mining. TKDE 2012.
>
> [3] Hashimoto, T. et al. Fairness without demographics in repeated loss minimization. In ICML 2018.
>
> [4] Lahoti, P. et al. Fairness without demographics through adversarially reweighted learning. In NeurIPS 2020.
>
> [5] Niu, S. et al. A decade survey of transfer learning (2010–2020). TAI 2020.
>
> [6] Tan, C. et al. A survey on deep transfer learning. In ICANN 2018.
>
> [7] Zhuang, F. et al. A comprehensive survey on transfer learning. Proceedings of the IEEE 2020.

---

> > ### Comment · Reviewer_V6Vx · 2022-08-09
> > **Thanks for responses**
> >
> > Thanks for the detailed response. I notice that the authors have answered questions I mentioned and claimed that they would add baseline and relevant works to their future version. I think expanding the discussion and explanation, plus the additional demonstration figure will improve the paper. However, the legibility and insight of the paper should be further improved.

---

> > > ### Author Response · Authors · 2022-08-09
> > > **Follow-up to Reviewer V6Vx**
> > >
> > > Thank you for the response.
> > >
> > > **Legibility:** We first empirically show the effectiveness of label smoothing through experiments on new Adult dataset in Tab. 1, and we theoretically discuss the connection between our knowledge distillation method for fairness and label smoothing in Section 3.4. We provide theoretical guarantee regarding EOd with promising empirical results showing improvement in terms of EOd under different demographics.
> > >
> > > **Insight:** Due to distributional disparities, fairness concerns exist without sensitive information, and improving fairness without demographics is even more difficult as it is hard to approximate fairness metrics without demographics. Existing works on fairness without demographics focus on Max-Min formulation; however, such formulation could be too strict to achieve improvement in group fairness. Instead, our work provides a **novel** perspective to address fairness without demographics through knowledge distillation, where we try to rectify the biased hard label with predicted soft label by teacher model. We hope our work can open new doors for addressing fairness concerns without demographics in many applications.

---

### Official Review · Reviewer_PRCV · 2022-07-11

**Rating:** 6
**Confidence:** 4
**Soundness:** 3 good
**Presentation:** 4 excellent
**Contribution:** 3 good

**Summary:**

The authors argue that training on distilled versions of existing classifiers improves fairness on unobserved demographics and build upon this objective. The approach is interesting and of use to the fairness literature, improving upon previous works. The paper is well-written, the experiment methodology is sound, and results convincing.

**Questions:**

Please see my comments on Strengths and Weaknesses section.

**Limitations:**

- Differences in Table 1 do not seem statistically significant in terms of disparate impact with a first glance due to high variance, so maybe a little more discussion is needed to address this (they are statistically significant for equalized odds though).
- In my opinion, the fact that the hyperparameter `a` can only be tweaked means that it is currently selected blindly. I think the authors need to be a little more explicit over this shortcoming. Maybe future works can try selecting that hyperparameter with few-shot tweaking (have sensitive attribute for a few samples and then reach an acceptable trade-off)?

**Strengths And Weaknesses:**

I have a few important concerns, though these are -in my opinion- straightforward to address:

- Given that the authors already arrived at Equation 9, why not use it directly for reweighting and then new training of the base model? I guess the only difference is whether new sample weights or new equivalent objectives are introduced, in which case I would like to see some discussion of this point in the text.
- The question of why disparate impact is improved is not really addressed. The improved theorization editing I mention in the following point could help address this, the authors can give a simple intuitive explanation / proposition, or ideally (though this could be too much work) a second theorem could be introduced. I remind that fairness is not a global objective and that efficacy over each sub-objective needs to be explored by its own. If none of the aforementioned actions are taken, I would also be happy seeing the paper emphasize more on improving equalized odds as the primary goal and mentioning that disparate impact is also improved as a side-effect (at least one hypothesis of why this holds true is needed - maybe something that starts with the assumption that parts of these biases could be correlated so partially fixing one also partially fixes the other and then working from there?).
- The authors argue in section 3.4 that the effectiveness of their approach can be attributed to down-weighting misclassified samples, which has been shown [Krasanakis et al., 2018] to be inherently insufficient by itself. As a recommendation on how to improve this (other explanations could be possible, perhaps even more intuitive depending on phrasing) with small adjustments to the text is to acknowledge that Equation 9 effectively introduces a different degree of weight skewing for highly misclassified samples (i.e. weights are not linear with respect to errors, since t(x) is also implicitly affected by errors) . So, protected demographics (i.e. the groups of people classifiers often bias against) are expected to obtain the highest misclassification rates and this is why placing disproportionately more weights on higher misclassifications addresses bias without impacting accuracy much.

Just a note:
It seems to me that Equation (9) is an ad-hoc variation of [Krasanakis et al., 2018]'s Equation 10 with some additional assumptions filling in for the absence of demographics. More analysis is needed to show equivalence for sure, and it is not needed by the paper's theorization, so I'm just mentioning this in case the authors find it interesting. If this holds true, there is a chance that tweaking the softmax's hyperparameter instead of selecting it with cross-validation could also create accuracy and fairness trade-offs, as already demonstrated for the hyperparmeter `a`, since the former implicitly affects the Lipshitz constant of sample weights.

Minor recommendations:
- in the abstract "Max-Min fairness objective." > "Max-Min fairness objectives."
- in the abstract "to achieve expected improvement in" > "to improve"
- I think the introduction needs a sentence to explain *why* the proposed approach is expected to trade-off fairness with accuracy.
- It would help adding [1] as a general background reference and [2] to labelling bias references in 3.2.
- If I understand this correctly, in section 3.2, probably explicitly mention that Binary corresponds to the original version of the Adult dataset.

[1] Ntoutsi, E., Fafalios, P., Gadiraju, U., Iosifidis, V., Nejdl, W., Vidal, M. E., ... & Staab, S. (2020). Bias in data‐driven artificial intelligence systems—An introductory survey. Wiley Interdisciplinary Reviews: Data Mining and Knowledge Discovery, 10(3), e1356.
[2] Iosifidis, Vasileios, and Eirini Ntoutsi. "Adafair: Cumulative fairness adaptive boosting." Proceedings of the 28th ACM International Conference on Information and Knowledge Management. 2019.

---

> ### Author Response · Authors · 2022-08-02
> **Response to reviewer PRCV**
>
> We thank the reviewer for the comment. We'll fix the typo and include the suggested reference in Introduction and Section 3.2 as suggested in final paper.
>
> * **Equivalence between reweighing and label smoothing:** We clarify that training student model with soft label $y'$ from knowledge distillation is equivalent to training the student model with binary labels under sample weights by Eq. (9).
>
> * **Fairness guarantee:** We show in Theorem 3.1 that our method can be directly related to equalized odds. Also, disparate impact and equalized odds can be related based on the following inequality (assume binary label and binary sensitive attribute):
> \begin{equation}
> \begin{aligned}
> DI& = |P[\hat{Y}=1|A=0] - P[\hat{Y}=1|A=1]|\\\\
> &=|P[\hat{Y}=1|Y=1,A=0]P[Y=1|A=0] + P[\hat{Y}=1|Y=0,A=0]P[Y=0|A=0] \\\\
> &\ \ \ \ - P[\hat{Y}=1|Y=1,A=1]P[Y=1|A=1]-P[\hat{Y}=1|Y=0,A=1]P[Y=0|A=1]|\\\\
> &\leq |P[\hat{Y}=1|Y=1,A=0]P[Y=1|A=0]-P[\hat{Y}=1|Y=1,A=1]P[Y=1|A=1]| \\\\
> &\ \ \ \ + |P[\hat{Y}=1|Y=0,A=0]P[Y=0|A=0]-P[\hat{Y}=1|Y=0,A=1]P[Y=0|A=1]|\\\\
> &=\alpha_0|P[\hat{Y}=1|Y=1,A=0]-\frac{\alpha_1}{\alpha_0}P[\hat{Y}=1|Y=1,A=1]| \\\\
> &+ (1-\alpha_0) |P[\hat{Y}=1|Y=0,A=0]-\frac{1-\alpha_1}{1-\alpha_0}P[\hat{Y}=1|Y=0,A=1]|\\\\
> &\leq max(\alpha_0,1-\alpha_0) \Big[ |P[\hat{Y}=1|Y=1,A=0]-\frac{\alpha_1}{\alpha_0}P[\hat{Y}=1|Y=1,A=1]|\\\\
> &\ \ \ \ \ \ \ \ \ \ \ \ +|P[\hat{Y}=1|Y=0,A=0]-\frac{1-\alpha_1}{1-\alpha_0}P[\hat{Y}=1|Y=0,A=1]| \Big],
> \end{aligned}
> \end{equation}
> Where $\alpha_i = P[Y=1|A=i]$. When the base rate ratio $\frac{\alpha_1}{\alpha_0}=1$, we have the upper bound same as equalized odds up to a multiplicative constant. If the discrepancy between base rates of different groups is not extremely large, it is possible to improve disparate impact by improving equalized odds, and smaller difference in base rate indicates that improvement in equalized odds and disparate impact could be achieved simultaneously. For example, the ratio between base rate of different **gender** on new Adult dataset is around $2.8$, while the ratio between base rate of different **race** is around $2$. As a result, improvement in equalized odds regarding sensitive attribute **race** benefits disparate impact more than improvement in equalized odds regarding sensitive attribute **gender** on new Adult dataset (as shown in Table 4, 5 of main paper). However, we do point out that when discrepancy between base rates of different groups are very large (for example, $\alpha_0=0.1$, $\alpha_1=0.9$), it is not feasible to improve both fairness metrics simultaneously.
>
> * **Effectiveness of our method:** The effectiveness of our approach is not merely attributed to down-weighting misclassified samples, which is studied in [Krasanakis et al., 2018]. As discussed in section 3.4, our method achieves two ideas: upweighting hard samples that are correctly classified and downweighting samples that are wrongly classified. In this way, we are paying more attention to part of hard samples, without focusing too much on wrongly classified hard samples.
>
> * **Differences in Table 1:** We perform t-test to validate the improvement in disparate impact by our linear label smoothing in Table 1. The baseline is chosen as "Binary" in Table 1 of main paper. The $p$-value for sensitive attribute _gender_ is 0.02, and the $p$-value for sensitive attribute _race_ is 0.11. This could be due to relatively larger variance of disparate impact regarding _race_ by our method. As discussed in main paper, our method focuses more on equalized odds, while disparate impact is more a side effect due to disparities in base rate (as discussed above in question **Fairness guarantee**).
>
> * **Few-shot tweaking:** We thank the reviewer for the kind suggestion. We first point out that our goal is to improve group fairness with no sensitive information available, similar to the settings of DRO [1] and ARL [2], and following similar ideas we fine-tune the hyperparameters to achieve _best classification performance_. Second, as the reviewer suggested, if we have partial sensitive information available, it is possible to fine-tune the hyperparameters to achieve _best fairness performance_ on the small set. However, we clarify that one potential problem regarding this idea could be that the partially available sensitive information might be likely to be biased, leading to a potentially unpromising fairness guarantee. We'll discuss this idea in future work.
>
> **[Reference]**
>
> [1] Hashimoto, T., Srivastava, M., Namkoong, H., \& Liang, P. Fairness without demographics in repeated loss minimization. In ICML 2018.
>
> [2] Lahoti, P., Beutel, A., Chen, J., Lee, K., Prost, F., Thain, N., ... \& Chi, E. Fairness without demographics through adversarially reweighted learning. In NeurIPS 2020.

---

### Author Response · Authors · 2022-08-05
**We are happy to answer further questions regarding our paper.**

Dear Reviewers,
Thank you for taking the time to review our work. We appreciate your detailed and constructive feeedback, and hopefully our reponse can address your concerns.
If you have more questions or conerns, please let us know. We would be happy to answer. Thank you very much.
Best,
Authors

---

### Meta-Review · Area_Chair_8tdf · 2022-08-24

**Recommendation:** Accept
**Confidence:** Certain

**Metareview:**

Overall the reviews are positive, leaning towards accept. The reviewers agree that the main idea is interesting and the paper is well written and structured. Also several issues raised are properly addressed and checked, and I think that the remaining issues are also taken care of. Hence, I recommend the acceptance of this paper.

**Award:**

No

---

### Decision · Program_Chairs · 2022-09-14

Accept